# HuR Promotes the Differentiation of Goat Skeletal Muscle Satellite Cells by Regulating Myomaker mRNA Stability

**DOI:** 10.3390/ijms24086893

**Published:** 2023-04-07

**Authors:** Yanjin Sun, Siyuan Zhan, Sen Zhao, Tao Zhong, Linjie Wang, Jiazhong Guo, Dinghui Dai, Dandan Li, Jiaxue Cao, Li Li, Hongping Zhang

**Affiliations:** 1Farm Animal Genetic Resources Exploration and Innovation Key Laboratory of Sichuan Province, Sichuan Agricultural University, Chengdu 611130, China; 2Key Laboratory of Livestock and Poultry Multi-Omics, Ministry of Agriculture and Rural Affairs, College of Animal and Technology, Sichuan Agricultural University, Chengdu 611130, China

**Keywords:** HuR, goat, Myomaker, MuSCs, stability

## Abstract

Human antigen R (HuR) is an RNA-binding protein that contributes to a wide variety of biological processes and diseases. HuR has been demonstrated to regulate muscle growth and development, but its regulatory mechanisms are not well understood, especially in goats. In this study, we found that HuR was highly expressed in the skeletal muscle of goats, and its expression levels changed during longissimus dorsi muscle development in goats. The effects of HuR on goat skeletal muscle development were explored using skeletal muscle satellite cells (MuSCs) as a model. The overexpression of HuR accelerated the expression of myogenic differentiation 1 (MyoD), Myogenin (MyoG), myosin heavy chain (MyHC), and the formation of myotubes, while the knockdown of HuR showed opposite effects in MuSCs. In addition, the inhibition of HuR expression significantly reduced the mRNA stability of MyoD and MyoG. To determine the downstream genes affected by HuR at the differentiation stage, we conducted RNA-Seq using MuSCs treated with small interfering RNA, targeting HuR. The RNA-Seq screened 31 upregulated and 113 downregulated differentially expressed genes (DEGs) in which 11 DEGs related to muscle differentiation were screened for quantitative real-time PCR (qRT-PCR) detection. Compared to the control group, the expression of three DEGs (Myomaker, CHRNA1, and CAPN6) was significantly reduced in the siRNA-HuR group (*p* < 0.01). In this mechanism, HuR bound to Myomaker and increased the mRNA stability of Myomaker. It then positively regulated the expression of Myomaker. Moreover, the rescue experiments indicated that the overexpression of HuR may reverse the inhibitory impact of Myomaker on myoblast differentiation. Together, our findings reveal a novel role for HuR in promoting muscle differentiation in goats by increasing the stability of Myomaker mRNA.

## 1. Introduction

Approximately 40 to 60 percent of an adult animal’s body weight is composed of skeletal muscle, making it the most abundant type of tissue in the body [1]. Skeletal muscle is composed of multinucleated contractile muscle cells (also called myofibers). During development, myofibers are formed by the fusion of progenitors from the mesoderm, known as myoblasts. In neonatal/juvenile stages, the number of myofibers remains constant, but each myofiber grows in size through the fusion of muscle satellite cells (MuSCs), a population of postnatal muscle stem cells [2,3]. In general, muscle satellite cells account for 30–35% of the sublaminal nuclei on myofibers in early postnatal murine muscles, and this number declines to 2–7% in adult muscles [2]. In adult skeletal muscle, MuSCs are quiescent but become active when skeletal muscle is damaged. When MuSCs become activated, they produce progeny and myoblasts and finally differentiate and fuse to form myotubes [4]. At the cellular level, myogenesis is controlled by the sequential expression of transcriptional regulators involving myogenic regulatory factors (MRFs), myocyte enhancer factor 2 (MEF2), and the Pax (paired box) family [5,6,7]. There is mounting evidence that long, noncoding RNAs (lncRNAs), microRNAs (miRNAs), and circular RNAs (circRNAs) play a role in the regulation of myogenesis [8,9,10]. It is worth mentioning that myogenesis is also controlled by the interaction between RNA binding proteins (RBPs) and coding and non-coding RNAs, which contributes to the growth and development of skeletal muscle satellite cells (MuSC) [11,12].

The RNA-binding protein HuR, also known as ELAV (embryonic lethal abnormal vision), like RNA-binding protein 1 (ELAV 1), is a crucial RNA binding protein that is widely expressed in various tissues of the body and plays key roles in several biological processes. In skeletal muscle, HuR plays a key role in the differentiation via stabilization of the mRNA transcripts of many important myogenic factors such as Myogenin, MyoD, and p21 [13,14]. During the early stages of myogenesis, HuR promotes the expression of the alarmin HMGB1 by preventing the miR-1192-mediated translation inhibition of its mRNA [15]. Simultaneously, HuR collaborates with the RBP KSRP (KH-type splicing regulatory protein) to reduce the expression of the nucleophosmin (NPM) protein by destabilizing its mRNA [16]. Linc-MD1 was the first lncRNA to be shown to play a relevant role in muscle differentiation by regulating specific myogenic factors required for the onset of late muscle gene transcription [17]. HuR regulates the expression of Linc-MD1 by favoring its accumulation in the cytoplasm at the expense of miR-133b synthesis, which is necessary for the correct progression of muscle differentiation [18]. In addition, the lncRNA OIP5-AS1 has been shown to bind to the 3′UTR of the myocyte-specific enhancer factor 2C (MEF2C) mRNA, whereas HuR binds both transcripts in myoblasts. HuR bound to MEF2C mRNA optimally only in the presence of OIP5-AS1. In turn, HuR binding to MEF2C mRNA results in the stabilization of MEF2C mRNA, increasing MEF2C levels and promoting myogenesis [19]. Another study found that HuR inhibition results in impaired metabolic flexibility and decreased lipid oxidation, suggesting a role for HuR as an important regulator of skeletal muscle metabolism [20]. Moreover, HuR counteracts miR-330 to promote STAT3 (signal transducer and activator of transcription 3) translation during inflammation-induced muscle wasting [21]. During the transition of myoblasts to myotubes, HuR associates with the YB1 protein in an RNA-independent manner. This complex is then recruited to a consensus motif in the 3′UTR of target mRNAs such as MyoG, MyoD, and c-Myc. As a result, the HuR/YB1 complex increases the stability of these mRNAs, resulting in the formation and maintenance of muscle fibers [22]. Despite several decades of study on HuR, our knowledge about its functions and mechanisms is still limited.

Here, we discovered that HuR is highly expressed in skeletal muscle tissues and is able to facilitate the differentiation of myoblasts effectively. Moreover, we identified the Myomaker as a novel HuR-binding partner during myogenic differentiation. Mechanistically, HuR binds to Myomaker and increases the mRNA stability of Myomaker, thereby activating MuSC differentiation by positively regulating the expression of Myomaker. In this study, we demonstrate that HuR is a crucial posttranslational regulator of muscle differentiation and identify a novel target of HuR in the promotion of muscle differentiation in goats.

## 2. Results

### 2.1. Expression Patterns of HuR

We performed a qRT-PCR to determine the HuR expression level in various goat tissues (lung, kidney, liver, brain, longissimus dorsi muscle, semitendinosus muscle, gastrocnemius muscle, psoas major muscle, adductor muscle, and semimembranosus muscle). The results showed that HuR is highly enriched in skeletal muscles (Figure 1A). In the longissimus dorsi muscle (LD), the expression level of HuR peaked on day 90 of gestation (Figure 1B). The muscle cell differentiation goat model was successfully constructed in order to detect the expression of HuR in goat muscle cells (Figure 1C,D). In addition, the expression levels of MyoG (marker of myogenic differentiation) during MuSC differentiation were quantified (Figure 1E). The expression of MyoG is consistent with a previous study [8] and demonstrated that HuR expression could be quantified accurately (Figure 1E). Furthermore, the expression levels of HuR during goat MuSC differentiation were measured, and the results indicated that the expression of HuR fluctuates during MuSC differentiation (Figure 1F).

### 2.2. HuR Promotes MuSC Differentiation in Goats

To address the function of HuR on the muscle differentiation, we conducted a functional gain/loss experiment on HuR in MuSCs. The results showed that the overexpression of HuR by the pEGFP-HuR vector significantly enhanced the mRNA expression of MyoG, MyoD, and MyHC (Figure 2A) and the protein abundance of MyHC (Figure 2B), whereas the knockdown of HuR had the opposite effect (Figure 2D,E). In addition, the immunofluorescence of MyHC showed that the overexpression of HuR resulted in the promotion of MuSC differentiation; this was assessed by an increased MyHC immunostaining and fusion index (Figure 2C). In contrast, the knockdown of HuR inhibits the differentiation of MuSCs, with a reduced MyHC immunostaining and fusion index (Figure 2F).

HuR promoted muscle differentiation by enhancing the mRNA stability of MyoD and MyoG in previous studies [23]. To monitor the effect of HuR on the stability of MyoD and MyoG mRNA after transfection with siRNA control and siRNA-HuR, we treated cultured MuSCs with actinomycin D (ActD, 5 µg/mL) to compromise the transcription process [24]. We extracted total RNA from MuSCs at 0, 1, 2, and 3 h after ActD treatment. We found that the stability of the MyoD and MyoG mRNA was significantly reduced in the siRNA-HuR group compared with the control group (Figure 2G,H). As the internal reference gene, the mRNA stability of GAPDH remained unchanged (Figure 2I). These results indicate that HuR promotes the differentiation of MuSCs by increasing the mRNA stability of MyoD and MyoG in goats.

### 2.3. Identification of HuR Downstream Targets

In order to systematically screen the downstream genes affected by HuR at the differentiation stage, we sequenced mRNA transcriptomes using siRNA-HuR and siRNA-control samples. First, the qRT-RCR results showed the successful knockdown of HuR in MuSCs after transfection with the siRNA-HuR and siRNA-control using ACTIN, GAPDH, and PGK1 as reference genes, respectively (Figure 3A). The samples were then used for mRNA-seq. A total of 144 differentially expressed genes (DEGs) were identified, including 31 upregulated and 113 downregulated DEGs (Figure 3B). Moreover, a GO enrichment analysis revealed that the DEGs were primarily involved in biological processes such as muscle structure development, muscle system process, and muscle organ development (Figure 3C). As a result of the KEGG analysis, the DEGs were found to be enriched in muscle-differentiation-related pathways, such as the PI3K-Akt signaling pathway and the MAPK signaling pathway (Figure 3D). A total of 11 DEGs related to muscle differentiation were screened for qRT-PCR detection. The results indicated that three DEGs (Myomaker, CHRNA1, and CAPN6) were significantly decreased in the HuR interference group compared with the control group (*p* < 0.01) (Figure 3E). It is noteworthy that Myomaker is a muscle-specific membrane protein that actively regulates the fusion of mononuclear myoblasts into multinucleated myofibers, as indicated in a previous study [25,26,27]. Thus, we performed RNA immunoprecipitation to explore the binding relationship between the HuR protein and Myomaker. Interestingly, we found that Myomaker was highly enriched by the HuR antibody compared with the control IgG (Figure 3F). These results imply that HuR most likely promotes muscle differentiation in goats by regulating Myomaker.

### 2.4. HuR Promotes the Differentiation of Goat MuSCs by Regulating Myomaker mRNA Stability

To explore whether Myomaker was regulated by HuR in goat MuSCs, we detected Myomaker mRNA and protein expression in MuSCs with HuR overexpression or HuR knockdown. The results showed that both Myomaker mRNA and protein were significantly increased when HuR was overexpressed (Figure 4A,B), while they were significantly decreased when HuR was knocked down (Figure 4C,D). To examine whether HuR regulated Myomaker expression in a post-transcription mechanism and in order to assess the effect of HuR on the stability of Myomaker mRNA, siRNA-control and siRNA-HuR were transfected into MuSCs differentiated for three days. The cultured MuSCs were then treated with actinomycin D (ActD, 5 µg/mL). The results of the qRT-PCR showed that the stability of the Myomaker mRNA was significantly reduced in the siRNA-HuR group (Figure 4E).

Furthermore, we performed co-transfection experiments using a HuR overexpression vector and Myomaker siRNA transfected into MuSCs differentiated for 3 days. The results showed that Myomaker knockdown decreased MyoG and MyHC expression levels. However, HuR overexpression abrogates this effect (Figure 4F,G). This indicates that HuR modulates the effect of Myomaker on MuSC differentiation. These results suggest that HuR promotes the differentiation of goat MuSCs by enhancing Myomaker stability.

## 3. Discussion

HuR plays critical roles in muscle development and disease. Previous studies have established that HuR promotes the differentiation of muscle by regulating the expression of myogenic regulators (p21, MyoD, MyoG). In addition, lncMGPF regulates myogenesis by enhancing the stability of HuR-mediated mRNAs such as MyoD and MyoG. It promotes muscle differentiation by facilitating HuR migration from the nucleus to the cytoplasm [28]. HuR promotes the transcription of the cardiac sodium channel gene (SCN5A) by binding to and enhancing the stability of MEF2C mRNA [29]. However, our knowledge of its role in muscle differentiation is still limited to goats. In the present study, we found that HuR expression was significantly higher in skeletal muscle than in other goat tissues, and expression changed during MuSC differentiation, suggesting that HuR might be involved in the regulation of muscle development. The effect of HuR overexpression and knockdown on MuSC differentiation was examined in vitro. The overexpression of HuR promoted muscle differentiation, whereas the inhibition of HuR had the opposite effect, as suggested by previous studies [13,14]. Several studies have demonstrated that HuR regulates mRNA expression either by stabilizing messages or by influencing their translation [30,31,32]. For instance, HuR bound to the 3′UTR of RAB5C, increasing RAB5C mRNA stability [33]. HuR affects FGFRL1 expression by binding to and stabilizing FGFRL1 mRNA [34]. Moreover, the mRNA stability of MyoG and MyoD decreased after interference with HuR, which is also consistent with previous findings [35]. Our results illustrated that HuR could promote the differentiation of goat MuSCs by enhancing MyoG and MyoD stability.

In order to systematically screen the downstream genes affected by HuR at the differentiation stage, we transfected siRNA-HuR into differentiating MuSCs and then performed mRNA transcriptome sequencing. A total of 144 differentially expressed genes (DEGs) were identified. Of these, 11 DEGs related to muscle differentiation were screened for qRT-PCR detection. Compared to the control group, the expression of three DEGs (Myomaker, CHRNA1, and CAPN6) was significantly reduced in the siRNA-HuR group (*p* < 0.01). It is noteworthy that Myomaker is a muscle-specific membrane protein that actively regulates the fusion of mononuclear myoblasts into multinucleated myofibers, as indicated in a previous study [25,26,27]. Interestingly, we found that Myomaker was highly enriched by the HuR antibody compared with the control IgG. These results suggest that HuR may promote muscle differentiation in goats by regulating Myomaker. The formation of skeletal muscle requires the mononucleated myoblasts to withdraw from the cell cycle and to fuse with each other to form nascent, multinucleated myotubes. As a result of further cell fusion, the nascent myotubes develop and express contractile proteins, which form mature myotubes. The fusion of myoblasts is a fundamental step during muscle differentiation, which involves a variety of cellular and molecular behaviors, including cell migration, recognition, adhesion, membrane alignment, signaling transduction, and actin cytoskeletal reorganization, leading to the final membrane fusion [36]. The fusion of myoblasts is an important step during skeletal muscle differentiation. Therefore, we speculate that HuR may regulate muscle differentiation by influencing the expression of Myomaker. Here, we found that Myomaker mRNA and protein were significantly increased when HuR was overexpressed, while they were significantly decreased when HuR was knocked down. Additionally, our results indicate that HuR positively regulates the stability of Myomaker mRNA. Myomaker, regulated by MyoD and MyoG, promotes chicken myoblast fusion, as previously reported [37]. In Myomaker knockout mice, MyoD and MyoG genes could be normally expressed, indicating that Myomaker does not affect the expression of differentiation marker genes [25]. In our study, we found that Myomaker knockdown decreased MyoG and MyHC expression; this may be caused by species differences.

In conclusion, our results demonstrate that HuR promotes the differentiation of goat MuSCs by enhancing Myomaker stability and identify a novel target of HuR in the promotion of muscle differentiation in goats.

## 4. Materials and Methods

### 4.1. Sample Preparation

The Animal Care and Use Committee of the College of Animal Science and Technology, Sichuan Agricultural University, Sichuan, China, approved all of the animal care, slaughter, and experimental procedures in accordance with the Regulations for the Administration of Affairs Concerning Experimental Animals (Ministry of Science and Tech-nology, Beijing, China) [SAU201418]. Pregnant goats (aged 2–3 years) were used in this study. Nine fetuses were removed through humane caesarean section at 90, 105, and 135 days of gestation (E90, E105, and E135). In addition, six female goats were sacrificed humanely on the third day after birth (B3) and 150 days after birth (B150). Longissimus dorsi muscle samples were obtained from these five developmental stages, and lung, kidney, liver, brain, semimembranosus muscle, semitendinosus muscle, gastrocnemius muscle, psoas major muscle, and adductor muscle samples were collected at B3. All samples were frozen in liquid nitrogen for RNA extraction.

### 4.2. Cell Culture and Transfection

Primary MuSCs were isolated and cultured from the longissimus dorsi (LD) muscle of a fetal goat (Chengdu Ma goat, female), as previously described [10,38]. MuSCs were seeded in 6-well (~2 × 10^5^ cells per well) or 12-well (~1 × 10^5^ cells per well) plates and culture in a growth medium (GM) consisting of Dulbecco’s Modified Eagle Medium (DMEM/high glucose, Meilunbio, Dalian, China) supplemented with 15% fetal bovine serum (FBS, Gibco, Grand Island, NY, USA). When the MuSCs density reached 80–90%, induced differentiation was carried out in a differentiation medium (DM) containing DMEM with 2% horse serum (HS, Gibco, Grand Island, NY, USA) and cultured in an incubator at 37 °C and 5% CO_2_.

For the gain and loss of function study, cells were transfected using Lipofectamine 3000 (Invitrogen, Waltham, MA, USA) with siRNAs or overexpression plasmids. After 8 h of transfection, the GM was replaced with DM. The transfected cells were harvested at 48 h (for RNA assay), 72 h (for protein assay), and immunofluorescence stained at the 4th day of differentiation.

### 4.3. Extraction of Total RNA and qRT-PCR

The total RNA was extracted using an RNAiso Plus reagent (TaKaRa, Dalian, China), according to the manufacturer’s instructions. RNA quality was assessed by electrophoresis on a 1.5% agarose gel, and RNA concentration was measured using a NanoDrop 2000 Spectrophotometer (Thermo-Fisher Scientific, Waltham, MA, USA). RNA (~1 µg) was reverse-transcribed into cDNA using the PrimeScriptTM RT Reagent Kit with gDNA Eraser (Takara, Dalian, China). A quantitative real-time PCR (qRT-PCR) was performed in a 10 µL volume containing 5 µL of ChamQ SYBR qRT-PCR Master Mix (Vazyme, Nanjing, China), 0.8 µL of cDNA, 3.4 µL of ddH_2_O, and 0.4 µL of each forward and reverse primer (10 µM), according to the manufacturer’s protocol. Each experiment was conducted independently three times with three biological replicates. The 2^−ΔΔCt^ method [39] was used to calculate the relative expression levels. As internal controls, the genes ACTB, GAPDH, and PGK1 were used. All primers used in this work are provided in Appendix A.

### 4.4. Plasmid Construction and siRNAs

To investigate the potential role of HuR, the pEGFP-N1 vector (Promega, Madison, WI, USA) was used to design the HuR overexpression vector using the HindIII and XhoI restriction sites. The primers used in vector construction are mentioned in Appendix A. Two small, interfering RNAs (siRNAs) against HuR and Myomaker were designed and synthesized by RiboBio (Guangzhou, China).

### 4.5. Western Blot Analysis

The antibodies used were MyHC (381620, ZENBIO), Myomaker (A18158, ABclonal), HuR (382170, ZENBIO), β-tubulin (200608, ZENBIO), and horseradish peroxidase (HRP)-conjugated anti-rabbit IgG (511203, ZENBIO). Total protein from in vitro cultured MuSCs was extracted using a total protein extraction kit (Solarbio, Beijing, China) and quantified using the BCA Protein Quantitation Kit (BestBio, Shanghai, China). In brief, ~20 µg of the qualified protein per samples were separated on a 10% SAS-PAGE, transferred to polyvinylidene fluoride (PVDF) membranes (Millipore, Burlington, MA, USA), blocked with 5% non-fat milk for 2 h at 37 °C, incubated with primary anti-rabbit for MyHC (1:500), Myomaker (1:1000), and HuR (1:1000) at 4 °C overnight and with a secondary antibody conjugated with HRP (1:10,000) for 1.5 h at 37 °C. Eventually, protein bands were exposed via the enhanced chemiluminescence detection system (BeyoECL Plus, TIANGEN, Beijing, China). β-tubulin antibody (1:1000) worked as a loading control.

### 4.6. Immunofluorescence Analyses

The MuSCs were transfected and cultured in DM. After 72 h, the culture medium was removed and the cells were washed three times with phosphate-buffered saline (PBS), fixed with 4% paraformaldehyde for 15 min at room temperature, washed three times again with 1 mL PBS after paraformaldehyde removal, permeabilized with 0.5% Triton X-100 for 10 min at 4°C, washed three times with 1 mL PBS, blocked with 1 mL of 2% bovine serum albumin (BSA) for 30 min at 37 °C, and incubated with anti-mouse MyHC (1:200; sc-376157, Santa Cruz, CA, USA) overnight at 4 °C and with secondary antibody IgG (H + L) (1:200, ABclonal, China) for 2 h at 37 °C. Finally, the cells were stained with 0.05 g/mL 4′,6′-diamidino-2-phenylindole (DAPI; Invitrogen) for 10 min at room temperature in the dark. Images were captured with a fluorescent inverted microscope (Leica, Wetzlar, Germany) and analyzed using ImageJ software. The fusion index was computed as the proportion of nuclei in fused myotubes with two or more nuclei. At least three samples were examined separately for each treatment.

### 4.7. RNA Stability Analyses

RNA stability was analyzed using the same methodology as described in previous studies [40]. Actinomycin D (ActD) was used to analyze the effect of HuR on Myomaker mRNA stability (ActD, Sigma, Shanghai, China). After treating the differentiated cells with ActD (5 µg /mL) for 48 h, we transfected MuSCs with siRNA-control or siRNA-HuR. Cells were harvested for qRT-PCR following transfection at 0, 1, 2, and 3 h.

### 4.8. RNA Immunoprecipitation Assay

The manufacturer’s instructions were followed for RNA immunoprecipitation (RIP), using the Magna RIP RNA-binding protein immunoprecipitation kit (Millipore, Billerica, MA, USA). In general, we collected cells that had differentiated for 5 days using a RIP lysis buffer. Then, 5 µg anti-HuR antibody (Abcam, Cambridge, UK) or an IgG (Millipore, Burlington, MA, USA) was mixed with magnetoglin A/G overnight at 4 °C for HuR immunoprecipitation. Finally, a qPCR analysis showed the abundance of Myomaker mRNA in the immunoprecipitated RNA.

### 4.9. RNA-Seq Analyses

Library preparation and poly(A) selection by RNA-seq were performed at Novogene Company (Beijing, China). In brief, the extraction of total RNA from transfected with siRNA-HuR or siRNA control in MuSCs (*n* = 3 per group). RNA integrity was assessed using the RNA Nano 6000 Assay Kit for the Bioanalyzer 2100 system. The NEBNext ^®^ Ultra ™ RNA Library Prep Kit was used for library preparation. Next, the Qubit2.0 Fluorometer, the Agilent 2100 Bioanalyzer, and a qRT-PCR were used to assess library quality. Then, qualified libraries were sequenced on the Illumina HiSeq 2500 platform (Illumina, San Diego, CA, USA) with a 2 × 150 bp pair-end [41].

Clean reads were obtained by removing the reads containing adapter, ploy-N, and low-quality reads from raw data. Then, DESeq2 (1.20.0) was used to analyze the differentially expressed genes (DEGs) between the two compared (|log2(Fold Change)| > 1 and padj < 0.05). Finally, we performed Gene Ontology (GO) and Kyoto Encyclopedia of Genes and Genomes (KEGG) analyses of the DEGs, which were implemented using the DESeq2 with padj < 0.05 and were considered significantly enriched [42].

### 4.10. Statistical Analysis

Data were the mean ± SEM, at least three biological replicates. GraphPad Prism 8.4 was used to conduct all statistical analyses. The unpaired two-tailed t-test was used for comparisons between two groups. * *p* < 0.05 and ** *p* < 0.01 represented statistical significance.

## Figures and Tables

**Figure 1 ijms-24-06893-f001:**
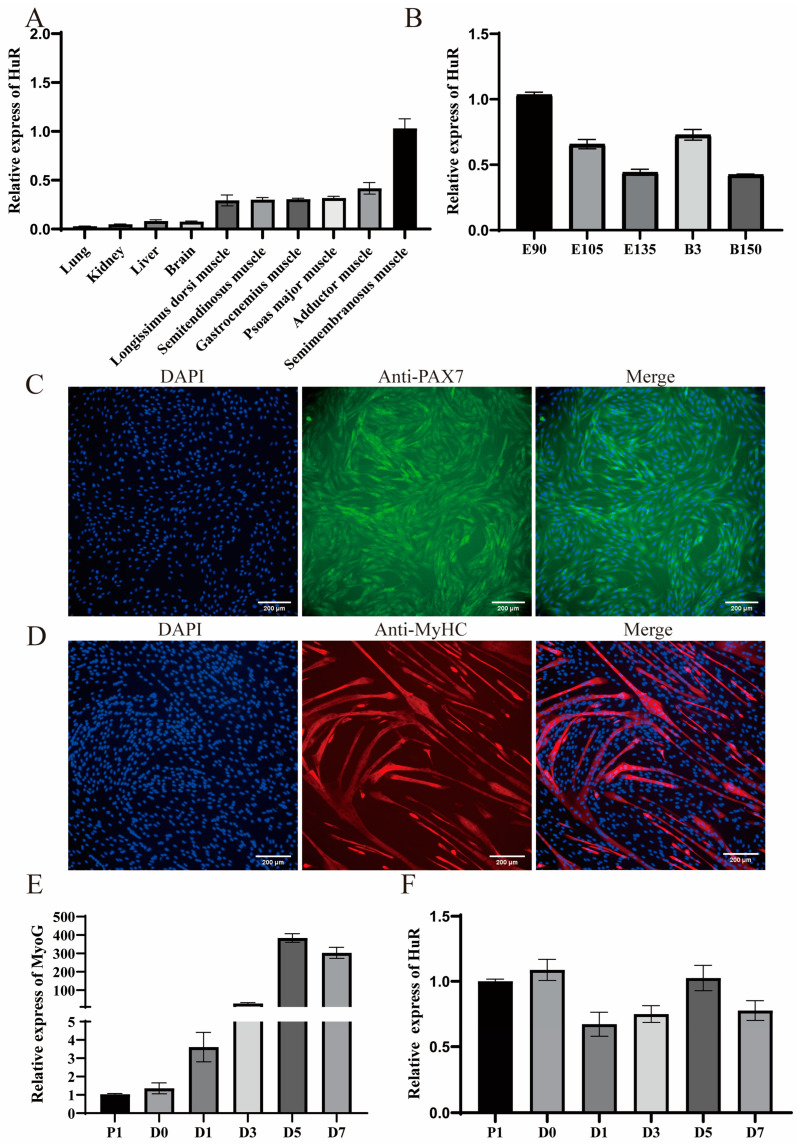
The expression profile of HuR. (**A**) qRT-PCR analysis of HuR expression in different tissues of goats on the third day after birth (B3). (**B**) The expression pattern of HuR in the developmental LD muscle. The embryonic period is defined as (**E**), and the postnatal period is defined as (**B**). (**C**) Representative immunofluorescence images of MuSCs stained using anti-Pax7 (green) and cultured in growth medium (GM). (**D**) MyHC immunofluorescence staining of MuSCs cultured in differentiation medium (DM) for 5 days. (**E**) The expression level of MyoG during MuSC differentiation (cultured in the growth medium (P1) and differentiation medium for 0, 1, 3, 5, and 7 days). (**F**) The expression pattern of HuR during differentiation of goat MuSCs (cultured in the growth medium (P1) and differentiation medium for 0, 1, 3, 5, and 7 days). Data are means ± standard error from at least three biological replicates.

**Figure 2 ijms-24-06893-f002:**
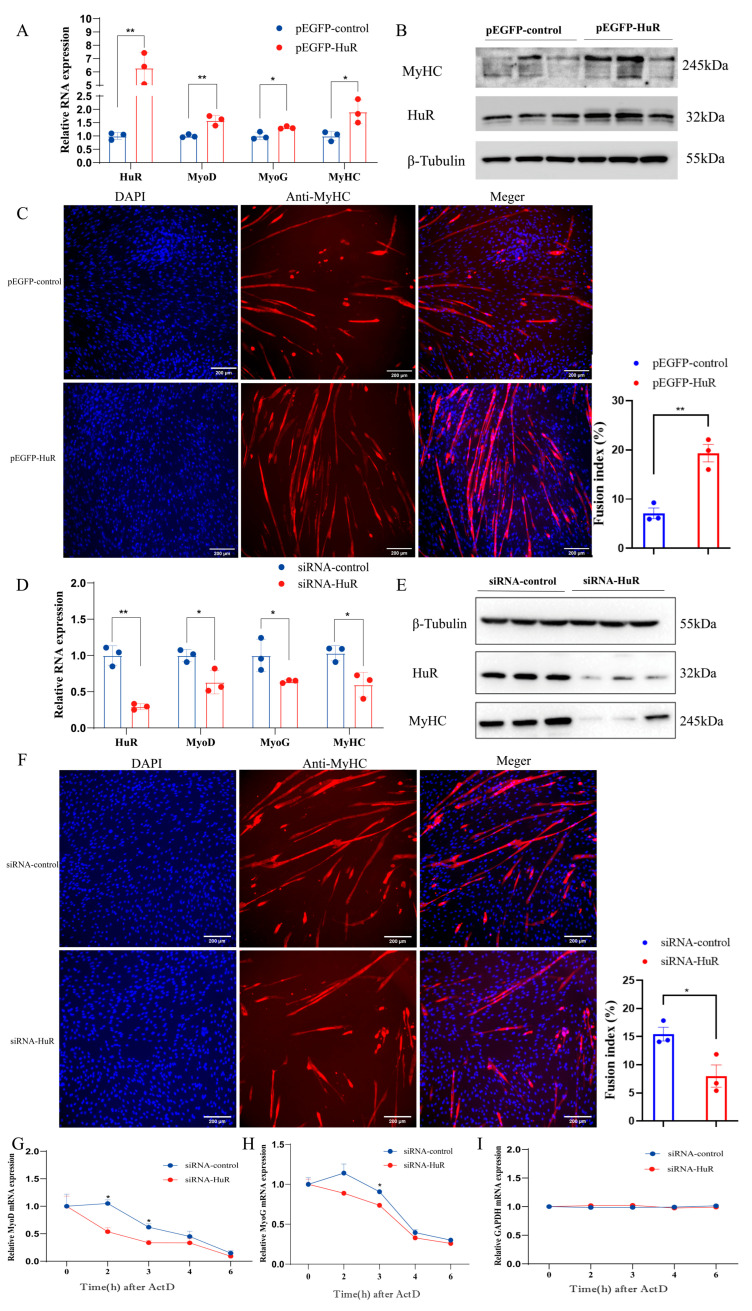
HuR promotes the differentiation of goat MuSCs. (**A**) The expression levels of HuR and muscle differentiation marker genes (MyoD, MyoG, and MyHC) were determined in MuSCs transfected with pEGFP-HuR or pEGFP-control. (**B**) MyHC protein levels were determined in MuSCs transfected with pEGFP-HuR or pEGFP-control. (**C**) Immunofluorescence detection of MyHC in MuSCs after 4 days of pEGFP-HuR transfection. DAPI labelling was used to view cell nuclei (blue). The fusion index of myotubes was calculated. Scale bars = 200 µm. (**D**) The expression levels of HuR and muscle differentiation marker genes (MyoD, MyoG, and MyHC) were determined in MuSCs transfected with siRNA-HuR or siRNA-control. (**E**) The protein levels of MyHC were determined in MuSCs transfected with siRNA-HuR or siRNA-control. (**F**) Immunofluorescence examination of MyHC-stained cells transfected with siRNA-HuR or siRNA-control for four days. Cell nuclei were visualized with DAPI (blue). The fusion index of myotubes was calculated. Scale bars = 200 µm. (**G**,**H**) HuR knockdown inhibits the stability of MyoD and MyoG mRNA. (**I**) HuR knockdown does not affect the stability of the internal reference gene GAPDH. The data are represented by the average of three independent experiments ± SEM; * *p* < 0.05; ** *p* < 0.01.

**Figure 3 ijms-24-06893-f003:**
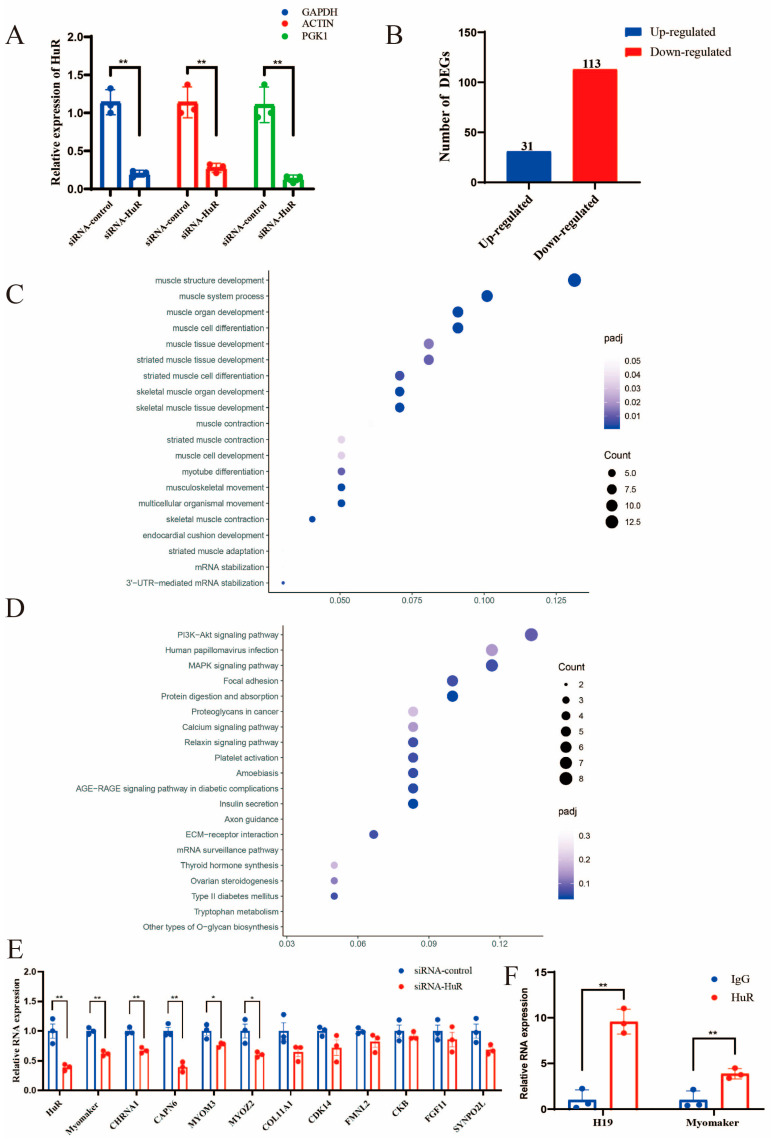
Identification of HuR downstream targets. (**A**) qRT-PCR showed that the expression of HuR in MuSCs transfected with siRNA-control or siRNA-HuR using different reference genes. (**B**) Numbers of upregulated and downregulated DEGs in goat MuSCs. (**C**) GO enrichment analysis of DEGs. (**D**) KEGG pathway enrichment analysis of DEGs. (**E**) qRT-PCR detection of 11 DEGs related to muscle differentiation. (**F**) RIP assay was carried out using HuR antibody, with IgG as the negative control. The H19 group served as a positive control. The data are represented by the average of three independent experiments ± SEM, * *p* < 0.05, ** *p* < 0.01.

**Figure 4 ijms-24-06893-f004:**
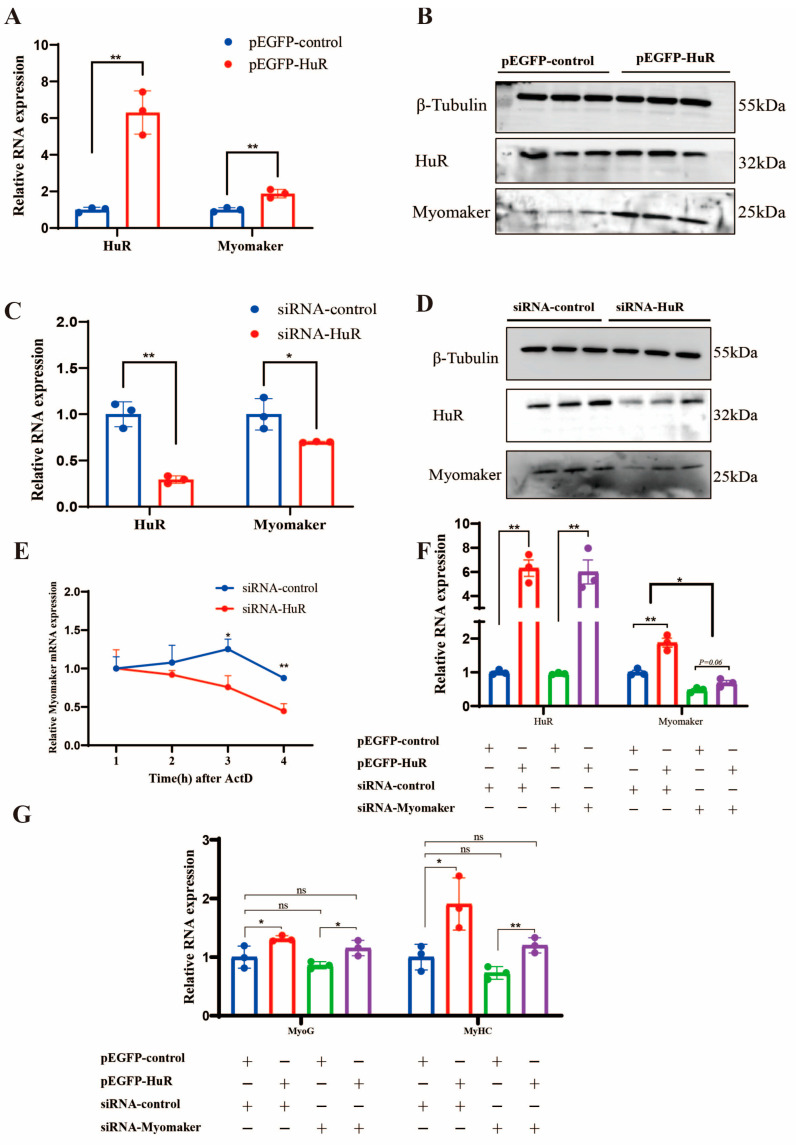
HuR promotes MuSC differentiation by modulating Myomaker stability. (**A**) The expression levels of HuR and Myomaker were determined in MuSCs transfected with pEGFP-HuR or pEGFP-control. (**B**) Myomaker protein levels were determined in MuSCs transfected with pEGFP-HuR or pEGFP-control. (**C**) The expression levels of HuR and Myomaker were determined in MuSCs transfected with siRNA-HuR or siRNA-control. (**D**) Myomaker protein levels were determined in MuSCs transfected with siRNA-HuR or siRNA-control. (**E**) qRT-PCR results showed that HuR knockdown inhibited the stability of the Myomaker mRNA. (**F**) The expression of HuR and Myomaker were determined after treatment with pEGFP-HuR and/or siRNA-Myomaker. (**G**) The expression of MyoG and MyHC was determined after treatment with pEGFP-HuR and/or siRNA-Myomaker. The data are represented by the average of three independent experiments ± SEM, * *p* < 0.05, ** *p* < 0.01.

## Data Availability

The accession number for the raw RNA sequencing data reported here is NCBI BioProject: PRJNA934189.

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
