# Peer review of "HuR Promotes the Differentiation of Goat Skeletal Muscle Satellite Cells by Regulating Myomaker mRNA Stability"

_ijms, 2023, doi:10.3390/ijms24086893_

Round 1

Reviewer 1 Report

The manuscript and results are interesting. Unfortunately, the quality of the text and description prevent the publication of the manuscript. English must be corrected by a native speaker. The language of the work is unscientific. Sentence constructions are incorrect and difficult to understand. The results are described carelessly. No description of Figures 2A and B. Figures 3C,D,F are completely illegible. The Authors in the the description of Figure 1 mentioned: "mean transmission electron microscope", but there are no electron microscope results on Fig 1. Which culture days and which media were studied Fig 2-4 ? The results are very poorly described. In order to reliably assess the content of the work, its form and language must be corrected.

Author Response

Dear Reviewer:

Thank you for your comments concerning our manuscript entitled “HuR Promotes the Differentiation of Goat Skeletal Muscle Satellite Cells by Regulating Myomaker mRNA Stability” (ID: ijms-2258904). Those comments are all valuable and very helpful for revising and improving our paper, as well as the important guiding significance to our research. We have carefully considered the comments and have made revisions to the manuscript point by point. We are very sorry for the mistakes in this manuscript and inconvenience they caused in your reading, the manuscript has been thoroughly revised and edited by a native speaker. We have made every effort to improve the quality and clarity of the language throughout the manuscript. In addition, some paragraphs have been rephrased as suggested. All the modifications are highlighted in red in the revised manuscript.

Please check the detailed responses and explanations listed below.

Best regards.

Yours sincerely,

Yanjin Sun

Farm Animal Genetic Resources Exploration and Innovation Key Laboratory of Sichuan Province, Sichuan Agricultural University, Chengdu 611130, China.

E-mail: s18098064595@163.com

  1. The manuscript and results are interesting. Unfortunately, the quality of the text and description prevent the publication of the manuscript. English must be corrected by a native speaker. The language of the work is unscientific. Sentence constructions are incorrect and difficult to understand.

Response: Thanks for your comments. We are very sorry for the mistakes in this manuscript and inconvenience they caused in your reading. The manuscript has been thoroughly revised and edited by a native speaker, and we have made every effort to improve the quality and clarity of the language throughout the manuscript. Thanks so much for your useful comments.

  1. The results are described carelessly.

Response: Thanks for your comments. We are very sorry for the carelessness in result section. The results have been rephrased as suggested. We have made a more rigorous and logical description of the results in the manuscript.

  1. No description of Figures 2A and B.

Response: Thanks for your comments. We are very sorry for our carelessness. We have added these descriptions of Figures 2A and 2B to the revised manuscript.

“The results showed that HuR overexpression by pEGFP-HuR vector significantly enhanced the mRNA expression of MyoG, MyoD, and MyHC (Figure 2A) and protein abundance of MyHC (Figure 2B), whereas knockdown of HuR had the opposite effect (Figure 2D, and 2E).”

  1. Figures 3C, D, F are completely illegible.

Response: Thanks for your comments. We have made corrections according to your comments. The details are shown in Figures 3C, 3D, and 3F.

  1. The Authors in the description of Figure 1 mentioned: "mean transmission electron microscope", but there are no electron microscope results on Fig 1.

Response: Thanks for your comments. We are very sorry for our negligence, and we have checked the manuscript carefully and found that this content does not match Figure 1. We have deleted this content from the revised manuscript.

  1. Which culture days and which media were studied Fig 2-4?

Response: Thanks for your comments. We have added this information to the cell culture and transfection sections.

“MuSCs were seeded in 6-well (~2 × 105 cells per well) or 12-well (~1 × 105 cells per well) plates and culture in the growth medium (GM) consisting of the Dulbecco's modified Eagle's medium (DMEM/high glucose, Meilunbio, Dalian, China) supplemented with 15% fetal bovine serum (FBS, Gibco, Grand Island, NY, USA). When the MuSCs density reaches 80-90%, induced differentiation is carried out in Differentiation medium (DM) containing DMEM with 2% horse serum (HS, Gibco, Grand Island, NY, USA) and cultured in an incubator at 37 °C and 5% CO2.

For the gain and loss of function study, cells were transfected using Lipofectamine 3000 (Invitrogen, Waltham, MA, USA) with siRNAs, or overexpression plasmids. After 8 h of transfection, GM was replaced with DM. The transfected cells were harvested at 48 h (for RNA assay), 72 h (for protein assay), and immunofluorescence stained at the 4th day of differentiation.”

  1. The results are very poorly described. In order to reliably assess the content of the work, its form and language must be corrected.

Response: Thanks for your comments. We are very sorry for the inconvenience in your reading. The results have been rephrased as suggested. The manuscript has been thoroughly revised and edited by a native speaker, and we have made every effort to improve the quality and clarity of the language throughout the manuscript.

Reviewer 2 Report

The paper proposed by Zhang et al. titled “HuR Promotes the Differentiation of Goat Skeletal Muscle Satellite Cells by Regulating Myomaker mRNA Stability” described the correlation among HuR RNA binding protein and the myogenic differentiation of Goat MuSC, identifying Myomaker RNA as HuR target to carry out its effect.

The presented results reveal different criticism, one related to myogenic capabilities of the goat MuSC, showing in some case 5-7% fusion while in some other more than 15% as shown in figure 2, hence it could be worth to standardize HuR effect in a more stable and coherent myogenic cell source.

In figure 3 is missing the panel G.

Recent publication showed that myogenic differentiation process is uncoupled with fusion, overall related to MyoD-Myomaker correlation. Thus, the results presented in the section 2.4 are quite contrasting with this new evidence showing myogenic differentiation and fusion as two complete separate events. So besides different results comparing already published research, the authors did not mention important literature about the topic and more important in the discussion they did not argue results discrepancies.

The manuscript needs a deep English revision, in several part the body text is remarkably unsmooth due to repetition in the sentences.

For these reasons I do not believe the present manuscript suitable for International Journal of Molecular Sciences  publication.

Author Response

Dear Reviewer:

Thank you for your comments concerning our manuscript entitled “HuR Promotes the Differentiation of Goat Skeletal Muscle Satellite Cells by Regulating Myomaker mRNA Stability” (ID: ijms-2258904). Those comments are all valuable and very helpful for revising and improving our paper, as well as the important guiding significance to our research. We have carefully considered the comments and have made revisions to the manuscript point by point. We are very sorry for the mistakes in this manuscript and inconvenience they caused in your reading, the manuscript has been thoroughly revised and edited by a native speaker. We have made every effort to improve the quality and clarity of the language throughout the manuscript. In addition, some paragraphs have been rephrased as suggested. All the modifications are highlighted in red in the revised manuscript.

Please check the detailed responses and explanations listed below.

Best regards.

Yours sincerely,

Yanjin Sun

Farm Animal Genetic Resources Exploration and Innovation Key Laboratory of Sichuan Province, Sichuan Agricultural University, Chengdu 611130, China.

E-mail: s18098064595@163.com

  1. The paper proposed by Zhang et al. titled “HuR Promotes the Differentiation of Goat Skeletal Muscle Satellite Cells by Regulating Myomaker mRNA Stability” described the correlation among HuR RNA binding protein and the myogenic differentiation of Goat MuSC, identifying Myomaker RNA as HuR target to carry out its effect.

Response: Thanks for your approval and suggestions. We have made corrections according to your comments.

  1. The presented results reveal different criticism, one related to myogenic capabilities of the goat MuSC, showing in some case 5-7% fusion while in some other more than 15% as shown in figure 2, hence it could be worth to standardize HuR effect in a more stable and coherent myogenic cell source.

Response: Thanks for your approval and suggestions. We agreed that the fusion differentiation of fig 2C and fig 2F. During our experiment, we separately transfected siRNA or pEGFP-N1 plasmid to differentiated MuSCs. Maybe, transfected pEGFP-N1 had a greater toxic effect on the cells, this is the primary reason why the fusion rate in fig 2C and fig 2F differs. In order to evaluate goat HuR function, we evaluated the fusion of cells transfected with overexpression (HuR) and inhibition (HuR) treatments, while the respective control groups served as standards. The fusion index was computed as the proportion of merged myotubes containing two or more nuclei.

  1. In figure 3 is missing the panel G.

Response: Thanks for your comments. We apologize for the mistake; Figure 3 only has 3A to 3F. we have made appropriate modifications in the revised draft.

  1. Recent publication showed that myogenic differentiation process is uncoupled with fusion, overall related to MyoD-Myomaker correlation. Thus, the results presented in the section 2.4 are quite contrasting with this new evidence showing myogenic differentiation and fusion as two complete separate events. So besides different results comparing already published research, the authors did not mention important literature about the topic and more important in the discussion they did not argue results discrepancies.

Response: Thanks for your comments. As a result of your suggestions, the discussion section of the revised manuscript has been rephrased.

“The formation of skeletal muscle requires the mononucleated myoblasts to withdraw from the cell cycle and to fuse with each other to form nascent, multinucleated myotubes. As a result of further cell fusion, the nascent myotubes develop and express contractile proteins, which form mature myotubes. The fusion of myoblasts is a fundamental step during muscle differentiation, which involves a variety of cellular and molecular behaviors, including cell migration, recognition, adhesion, membrane alignment, signaling transduction, and actin cytoskeletal reorganization, leading to the final membrane fusion [1]. The fusion of myoblasts is an important step during skeletal muscle differentiation. Therefore, we speculate that HuR may regulate muscle differentiation by influencing the expression of Myomaker. Here, we found that Myomaker mRNA and protein were significantly increased when HuR was overexpressed, while they were significantly decreased when HuR was knocked down. Additionally, our results indicated that HuR positively regulates the stability of Myomaker mRNA. Myomaker, regulated by MyoD and MyoG, promotes chicken myoblast fusion, as previously reported [2]. In Myomaker knockout mice, MyoD and MyoG genes could be normally expressed, indicating that Myomaker does not affect the expression of differentiation marker genes [3]. In our study, we found that Myomaker knockdown decreased MyoG and MyHC expression, this may be caused by species differences.”

  1. The manuscript needs a deep English revision, in several part the body text is remarkably unsmooth due to repetition in the sentences.

Response: Thanks for your comments. We are very sorry for the mistakes in this manuscript and inconvenience they caused in your reading. The manuscript has been thoroughly revised and edited by a native speaker, and we have made every effort to improve the quality and clarity of the language throughout the manuscript.

  1. For these reasons I do not believe the present manuscript suitable for International Journal of Molecular Sciences publication.

Response: Thank you so much for reviewing our paper and giving us valuable comments. We have studied comments carefully and have made corrections which we hope meet with approval.

  1. Abmayr, S.M.; Balagopalan, L.; Galletta, B.J.; Hong, S.J. Cell and molecular biology of myoblast fusion. Int Rev Cytol 2003, 225, 33-89, doi:10.1016/s0074-7696(05)25002-7.
  2. Luo, W.; Li, E.; Nie, Q.; Zhang, X. Myomaker, Regulated by MYOD, MYOG and miR-140-3p, Promotes Chicken Myoblast Fusion. Int J Mol Sci 2015, 16, 26186-26201, doi:10.3390/ijms161125946.
  3. Millay, D.P.; O'Rourke, J.R.; Sutherland, L.B.; Bezprozvannaya, S.; Shelton, J.M.; Bassel-Duby, R.; Olson, E.N. Myomaker is a membrane activator of myoblast fusion and muscle formation. Nature 2013, 499, 301-305, doi:10.1038/nature12343.

Reviewer 3 Report

Sun et al. found the RNA-binding protein HuR promotes muscle stem cell differentiation in goats and identified a downstream target gene Myomaker. Since it has been well studied that HuR promotes myogenic differentiation, this paper tried to find new targets of HuR.

However, there are quite a few points of concern to question the reliability of the study.

1.     Line 279. “MuSCs were extracted from the longissimus dorsi of 90-day-old Chengdu hemp sheep embryos … ” The authors studied muscle stem cells of goat (title and introduction), but used sheep to isolate muscle stem cells. Goat and sheep are two different species. In addition, it’s not clear if the authors used 90-day-old sheep, or used sheep embryos.

2.     It’s not clear how the authors isolate and purify muscle stem cells from goat. Immunostaining of PAX7 should be provided to identify muscle stem cells. If using FACS to isolate muscle stem cells, cell surface markers should be indicated.

3.     Line 281. How can the authors only use DMEM to culture muscle stem cells?

4.     Figure 1B, which tissue usde is not indicated.

5.     Figure 1C and 1E, what P1 and P2 represented for?

6.     Line 111. “… transmission electron microscope …”, what’s this?

7.     Line 116. “… scanning electron microscope…”, what’s this?

8.     Figure 2C, upper left and upper right image does not match.

9.     Figure 3. Cannot find figure 3G.

10. RNA-seq is the most informative method is this study, but there is no detailed information about RNA-seq - which cells are used, how many samples, how the RNA-seq analysis is performed.

11.  Line 199 to Line 202, and Figure 4G. The figure 4G didn’t suggest Myomaker knockdown downregulated MyoD and MyoG mentioned in Line 199. The descriptions “… the suppression of MuSCs …“ and “… muscle mass…” are not related to the figure 4. Figure legend of figure 4G is missing.

12.  The Discussion part didn’t discuss the results of this study at all.

13.  The English of this manuscript is very difficult to understand. 

Author Response

Dear Reviewer::

Thank you for your comments concerning our manuscript entitled “HuR Promotes the Differentiation of Goat Skeletal Muscle Satellite Cells by Regulating Myomaker mRNA Stability” (ID: ijms-2258904). Those comments are all valuable and very helpful for revising and improving our paper, as well as the important guiding significance to our research. We have carefully considered the comments and have made revisions to the manuscript point by point. We are very sorry for the mistakes in this manuscript and inconvenience they caused in your reading, the manuscript has been thoroughly revised and edited by a native speaker. We have made every effort to improve the quality and clarity of the language throughout the manuscript. In addition, some paragraphs have been rephrased as suggested. All the modifications are highlighted in red in the revised manuscript.

Please check the detailed responses and explanations listed below.

Best regards.

Yours sincerely,

Yanjin Sun

Farm Animal Genetic Resources Exploration and Innovation Key Laboratory of Sichuan Province, Sichuan Agricultural University, Chengdu 611130, China.

E-mail: s18098064595@163.com

Sun et al. found the RNA-binding protein HuR promotes muscle stem cell differentiation in goats and identified a downstream target gene Myomaker. Since it has been well studied that HuR promotes myogenic differentiation, this paper tried to find new targets of HuR.

However, there are quite a few points of concern to question the reliability of the study.

Response: Thanks for your approval and suggestions. We have made corrections according to your comments.

  1. Line 279. “MuSCs were extracted from the longissimus dorsi of 90-day-old Chengdu hemp sheep embryos …” The authors studied muscle stem cells of goat (title and introduction), but used sheep to isolate muscle stem cells. Goat and sheep are two different species. In addition, it’s not clear if the authors used 90-day-old sheep, or used sheep embryos.

Response: Thanks for your comments. We are sorry for our carelessness that the incorrect description of experimental animals. We have made corrections in revised manuscript.

“Primary MuSCs were isolated and cultured from the longissimus dorsi (LD) muscle of a fetal goat (Chengdu Ma goat, female)….”

  1. It’s not clear how the authors isolate and purify muscle stem cells from goat. Immunostaining of PAX7 should be provided to identify muscle stem cells. If using FACS to isolate muscle stem cells, cell surface markers should be indicated.

Response: Thanks for your comments. We have added this information in response to your comments.

“Primary MuSCs were isolated and cultured from the longissimus dorsi (LD) muscle of a fetal goat (Chengdu Ma goat, female), as previously described [1,2].”

Briefly, after a quick washing step with sterile phosphate buffered saline (PBS, Hyclone), sampled LD muscles were minced and digested with 0.2% Pronase (Sigma-Aldrich) at 37°C for 1h, followed by centrifuging at 1500×g for 6min and then pellet was kept. After further washed twice with PBS, the pellets were suspended in Dulbecco's modified Eagle's medium (DMEM/high glucose, Hyclone) supplemented 15% fetal bovine serum (FBS, Gibco), filtered through a 70-μm-mesh sieve (BD) and then span at 800 ×g for 5min to isolate the MuSCs. The isolated MuSCs were purified using a Percoll gradient (90, 40, and 20%) (Sigma-Aldrich) and centrifuged at 1800 ×g for 50min to enrich SMSCs in the Percoll interface between 40% and 90%. Finally, the purified MuSCs were qualified by directly immunostaining with Pax7 (Paired box 7, rabbit anti-Pax7, 1:100 dilution, Boster), a critical marker for MuSCs. Those Pax7+ SMSCs were kept in liquid nitrogen for the next experiments.

[1] Li, L.; Chen, Y.; Nie, L.; Ding, X.; Zhang, X.; Zhao, W.; Xu, X.; Kyei, B.; Dai, D.; Zhan, S.; et al. MyoD-induced circular RNA CDR1as promotes myogenic differentiation of skeletal muscle satellite cells. Biochim Biophys Acta Gene Regul Mech 2019, 1862, 807-821, doi:10.1016/j.bbagrm.2019.07.001.

[2] Zheng, S.; Li, L.; Zhou, H.; Zhang, X.; Xu, X.; Dai, D.; Zhan, S.; Cao, J.; Guo, J.; Zhong, T.; et al. CircTCF4 Suppresses Proliferation and Differentiation of Goat Skeletal Muscle Satellite Cells Independent from AGO2 Binding. Int J Mol Sci 2022, 23, doi:10.3390/ijms232112868.

  1. Line 281. How can the authors only use DMEM to culture muscle stem cells?

Response: Thanks for your comments. We are very sorry for our negligence, and we have checked the manuscript carefully and made corrections according to your comments.

“MuSCs were seeded in 6-well (~2 × 105 cells per well) or 12-well (~1 × 105 cells per well) plates and culture in the growth medium (GM) consisting of the Dulbecco's modified Eagle's medium (DMEM/high glucose, Meilunbio, Dalian, China) supplemented with 15% fetal bovine serum (FBS, Gibco, Grand Island, NY, USA). When the MuSCs density reaches 80-90%, induced differentiation is carried out in Differentiation medium (DM) containing DMEM with 2% horse serum (HS, Gibco, Grand Island, NY, USA) and cultured in an incubator at 37 °C and 5% CO2.”

  1. Figure 1B, which tissue used is not indicated.

Response: Thanks for your comments. We are sorry for our carelessness. We have added this information in Figure 1B.

“(B) The expression pattern of HuR in the developmental LD muscle.”

  1. Figure 1C and 1E, what P1 and P2 represented for?

Response: Thanks for your comments. P1 and P2 represent a proliferation density of 50% and 80%, respectively.

  1. Line 111. “… transmission electron microscope …”, what’s this?

Response: Thanks for your comments. We are very sorry for our negligence, and we have checked the manuscript carefully and found that this content does not match fig 1. Now we have deleted this content in line 111.

  1. Line 116. “… scanning electron microscope…”, what’s this?

Response: Thanks for your comments. We are very sorry for our negligence, and we have checked the manuscript carefully and found that this content does not match Figure 1. We have deleted this content from the revised manuscript.

  1. Figure 2C, upper left and upper right image does not match.

Response: Thanks for your comments. We are very sorry for our negligence, and we have made corrections in Figure 2C according to your comments.

  1. Figure 3. Cannot find figure 3G.

Response: Thanks for your comments. We apologize for the mistake; Figure 3 only has 3A to 3F. we have made appropriate modifications in the revised draft.

  1. RNA-seq is the most informative method is this study, but there is no detailed information about RNA-seq - which cells are used, how many samples, how the RNA-seq analysis is performed.

Response: Thanks for your comments. We are sorry for our negligence. We have added this information in Materials and Methods.

  1. Line 199 to Line 202, and Figure 4G. The figure 4G didn’t suggest Myomaker knockdown downregulated MyoD and MyoG mentioned in Line 199. The descriptions “… the suppression of MuSCs …“and “… muscle mass…” are not related to the figure 4. Figure legend of figure 4G is missing.

Response: Thanks for your comments. We are very sorry for the inconvenience in your reading. The sections have been rephrased and information about figure 4G has been added to the figure legend.

“The results showed that Myomaker knockdown decreased MyoG and MyHC expression levels. However, HuR overexpression abrogates this effect (Figure 4F, 4G). This indicates that HuR modulates the effect of Myomaker on MuSCs differentiation. These results suggest that HuR promotes the differentiation of goat MuSCs by enhancing Myomaker stability.”

“(G) The expression of MyoG and MyHC were determined after treatment with pEGFP-HuR and/or siRNA-Myomaker.”

  1. The Discussion part didn’t discuss the results of this study at all.

Response: Thanks for your comments. As a result of your suggestions, the discussion section of the revised manuscript has been rephrased.

  1. The English of this manuscript is very difficult to understand.

Response: Thanks for your comments. We are very sorry for the mistakes in this manuscript and inconvenience they caused in your reading. The manuscript has been thoroughly revised and edited by a native speaker, and we have made every effort to improve the quality and clarity of the language throughout the manuscript.

Round 2

Reviewer 1 Report

The paper could be accepted in present form. However,editing the text would improve its quality.

Author Response

Dear Reviewer:

Thank you again for your comments concerning our manuscript entitled “HuR Promotes the Differentiation of Goat Skeletal Muscle Satellite Cells by Regulating Myomaker mRNA Stability” (ID: ijms-2258904). Those comments are all valuable and very helpful for revising and improving our paper, as well as the important guiding significance to our research. We have carefully considered the comments and have made revisions to the manuscript point by point. We further have made every effort to improve the quality and clarity of the language throughout the manuscript. All the modifications are highlighted in red in the revised manuscript.

Please check the detailed responses and explanations listed below.

Best regards.

Yours sincerely,

Yanjin Sun

Farm Animal Genetic Resources Exploration and Innovation Key Laboratory of Sichuan Province, Sichuan Agricultural University, Chengdu 611130, China.

E-mail: s18098064595@163.com

Comments and Suggestions for Authors

The paper could be accepted in present form. However, editing the text would improve its quality.

Response: Thank you again for your constructive comments during the review process. we again revised and edited the manuscript and made every effort to improve the quality and clarity of language throughout the manuscript. Thank you again for your affirmation of the manuscript, we hope meet with approval.

Reviewer 3 Report

The revised version looks good to me.

Several minor points:

1.      Fig 1B,E,F: Figure legend should indicate the meaning of all those abbreviations, such as E90, P1.

2.      Line 118, how the fusion index is measured should be included in the methods.

3.      Fig 3B, the title of y axis should be “Number of DEGs”.

4.      Fig 4G, to conclude that “Myomaker knockdown decreased MyoG and MyHC expression levels. However, HuR overexpression abrogates this effect”, the comparisons between blue bar vs. green bar, and comparisons between blue bar vs. purple bar, should be indicated (significant or n.s.) within the MyoG group and within the MyHC group.

5.      If possible, please include in the introduction some background of goat muscle and goat muscle satellite cells, e.g., what proportion of mononuclear cells are satellite cells in goat muscle? This will help readers to understand why you study goat satellite cells.

Author Response

Dear Reviewer:

Thank you again for your comments concerning our manuscript entitled “HuR Promotes the Differentiation of Goat Skeletal Muscle Satellite Cells by Regulating Myomaker mRNA Stability” (ID: ijms-2258904). Those comments are all valuable and very helpful for revising and improving our paper, as well as the important guiding significance to our research. We have carefully considered the comments and have made revisions to the manuscript point by point. We further have made every effort to improve the quality and clarity of the language throughout the manuscript. All the modifications are highlighted in red in the revised manuscript.

Please check the detailed responses and explanations listed below.

Best regards.

Yours sincerely,

Yanjin Sun

Farm Animal Genetic Resources Exploration and Innovation Key Laboratory of Sichuan Province, Sichuan Agricultural University, Chengdu 611130, China.

E-mail: s18098064595@163.com

Comments and Suggestions for Authors

The revised version looks good to me.

Response: Thank you again for your positive comments during the review process. We have made corrections according to your comments.

Several minor points:

  1. Fig 1B,E,F: Figure legend should indicate the meaning of all those abbreviations, such as E90, P1.

Response: Thanks for your comments. We have added this information in the figure legend to your comments.

“Fig 1B, The expression pattern of HuR in the developmental LD muscle. The embryonic period is defined as E, and the postnatal period is defined as B.“

“Fig 1E, The expression level of MyoG during MuSC differentiation (cultured in the growth medium (P1) and differentiation medium for 0, 1, 3, 5, and 7 days).“

“Fig 1F, The expression pattern of HuR during differentiation of goat MuSCs (cultured in the growth medium (P1) and differentiation medium for 0, 1, 3, 5, and 7 days).“

  1. Line 118, how the fusion index is measured should be included in the methods.

Response: Thanks for your comments. We have added this information in Materials and Methods (lines 351-353).

“The fusion index was computed as the proportion of nuclei in fused myotubes with two or more nuclei. At least three samples were examined separately for each treatment. “

  1. Fig 3B, the title of y axis should be “Number of DEGs”.

Response: Thanks for your comments. We are very sorry for our negligence, and we have made corrections in Figure 3B according to your comments.

  1. Fig 4G, to conclude that “Myomaker knockdown decreased MyoG and MyHC expression levels. However, HuR overexpression abrogates this effect”, the comparisons between blue bar vs. green bar, and comparisons between blue bar vs. purple bar, should be indicated (significant or n.s.) within the MyoG group and within the MyHC group.

Response: Thanks for your comments. We are very sorry for our negligence, and we have made corrections in Figure 4G according to your comments.

  1. If possible, please include in the introduction some background of goat muscle and goat muscle satellite cells, e.g., what proportion of mononuclear cells are satellite cells in goat muscle? This will help readers to understand why you study goat satellite cells.

Response: Thanks for your comments. As a result of your suggestions, we have added some background of goat muscle and goat muscle satellite cells in the introduction.

“Approximately 40 to 60 percent of an adult animal's body weight is composed of skeletal muscle, making it the most abundant type of tissue in the body [1]. Skeletal muscle is composed of multinucleated contractile muscle cells (also called myofibers). During development, myofibers are formed by the fusion of progenitors from the mesoderm, known as myoblasts. In neonatal/juvenile stages, the number of myofibers remains constant, but each myofiber grows in size by fusion of muscle satellite cells (MuSCs), a population of postnatal muscle stem cells [2,3]. In general, muscle satellite cells account for 30-35% of the sublaminal nuclei on myofibers in early postnatal murine muscles, and this number declines to 2-7% in adult muscles [2]. In adult skeletal muscle, MuSCs are quiescent, but become active when skeletal muscle is damaged. When MuSCs become activated, they produce progeny, myoblasts, and finally differentiate and fuse to form myotubes [4]. At the cellular level, myogenesis is controlled by sequential expression of transcriptional regulators involving myogenic regulatory factors (MRFs), myocyte enhancer factor 2 (MEF2), and the Pax (paired box) family [5-7]. There is mounting evidence that long noncoding RNAs (lncRNAs), microRNAs (miRNAs), and circular RNAs (circRNAs) play a role in the regulation of myogenesis [8-10]. It is worth mentioning that myogenesis is also controlled by the interaction between RNA binding proteins (RBPs) and coding and non-coding RNAs, which contributes to the growth and development of skeletal muscle satellite cells (MuSC) [11,12].”

  1. Khanna, S.; Merriam, A.P.; Gong, B.; Leahy, P.; Porter, J.D. Comprehensive expression profiling by muscle tissue class and identification of the molecular niche of extraocular muscle. Faseb j 2003, 17, 1370-1372, doi:10.1096/fj.02-1108fje.
  2. Yin, H.; Price, F.; Rudnicki, M.A. Satellite cells and the muscle stem cell niche. Physiol Rev 2013, 93, 23-67, doi:10.1152/physrev.00043.2011.
  3. Bentzinger, C.F.; Wang, Y.X.; Rudnicki, M.A. Building muscle: molecular regulation of myogenesis. Cold Spring Harb Perspect Biol 2012, 4, doi:10.1101/cshperspect.a008342.
  4. Yamanouchi, K.; Hosoyama, T.; Murakami, Y.; Nakano, S.; Nishihara, M. Satellite cell differentiation in goat skeletal muscle single fiber culture. J Reprod Dev 2009, 55, 252-255, doi:10.1262/jrd.20175.
  5. Bhagavati, S.; Song, X.; Siddiqui, M.A. RNAi inhibition of Pax3/7 expression leads to markedly decreased expression of muscle determination genes. Mol Cell Biochem 2007, 302, 257-262, doi:10.1007/s11010-007-9444-3.
  6. Taylor, M.V.; Hughes, S.M. Mef2 and the skeletal muscle differentiation program. Semin Cell Dev Biol 2017, 72, 33-44, doi:10.1016/j.semcdb.2017.11.020.
  7. Ott, M.O.; Bober, E.; Lyons, G.; Arnold, H.; Buckingham, M. Early expression of the myogenic regulatory gene, myf-5, in precursor cells of skeletal muscle in the mouse embryo. Development 1991, 111, 1097-1107, doi:10.1242/dev.111.4.1097.
  8. Zhan, S.; Qin, C.; Li, D.; Zhao, W.; Nie, L.; Cao, J.; Guo, J.; Zhong, T.; Wang, L.; Li, L.; et al. A Novel Long Noncoding RNA, lncR-125b, Promotes the Differentiation of Goat Skeletal Muscle Satellite Cells by Sponging miR-125b. Front Genet 2019, 10, 1171, doi:10.3389/fgene.2019.01171.
  9. Zhao, W.; Yang, H.; Li, J.; Chen, Y.; Cao, J.; Zhong, T.; Wang, L.; Guo, J.; Li, L.; Zhang, H. MiR-183 promotes preadipocyte differentiation by suppressing Smad4 in goats. Gene 2018, 666, 158-164, doi:10.1016/j.gene.2018.05.022.
  10. Li, L.; Chen, Y.; Nie, L.; Ding, X.; Zhang, X.; Zhao, W.; Xu, X.; Kyei, B.; Dai, D.; Zhan, S.; et al. MyoD-induced circular RNA CDR1as promotes myogenic differentiation of skeletal muscle satellite cells. Biochim Biophys Acta Gene Regul Mech 2019, 1862, 807-821, doi:10.1016/j.bbagrm.2019.07.001.
  11. Zhou, G.; Yang, Y.; Zhang, X.; Wang, J. Msx1 cooperates with Runx1 for inhibiting myoblast differentiation. Protein Expr Purif 2021, 179, 105797, doi:10.1016/j.pep.2020.105797.
  12. Shi, D.L.; Grifone, R. RNA-Binding Proteins in the Post-transcriptional Control of Skeletal Muscle Development, Regeneration and Disease. Front Cell Dev Biol 2021, 9, 738978, doi:10.3389/fcell.2021.738978.
